# Xylem Transcriptome Analysis in Contrasting Wood Phenotypes of *Eucalyptus urophylla* × *tereticornis* Hybrids

Xianliang Zhu [1,†] , Jiayue He [1,†], Changpin Zhou [1], Qijie Weng [1], Shengkan Chen [2], David Bush [3]
and Fagen Li [1,*]

1   Key Laboratory of National Forestry and Grassland Administration on Tropical Forestry Research, Research Institute of Tropical Forestry, Chinese Academy of Forestry, Guangzhou 510520, China; xianliangzhu2021@126.com (X.Z.); ssmilsy@outlook.com (J.H.); zhouchangpin@126.com (C.Z.); qjw65@126.com (Q.W.)
2   Guangxi Key Laboratory of Superior Timber Trees Resource Cultivation & Key Laboratory of Central South Fast-Growing Timber Cultivation of Forestry Ministry of China, Guangxi Forestry Research Institute, Nanning 530002, China; chenshengkan@126.com
3   CSIRO Australian Tree Seed Centre, GPO Box 1600, Canberra, ACT 2601, Australia; david.bush@csiro.au
*   Correspondence: lifagen@caf.ac.cn
†   These authors contributed equally to this work.

**Abstract:** An investigation of the effects of two important post-transcriptional regulatory mechanisms, gene transcription and alternative splicing (AS), on the wood formation of *Eucalyptus urophylla × tereticornis*, an economic tree species widely planted in southern China, was carried out. We performed RNA-seq on *E. urophylla × tereticornis* hybrids with highly contrasting wood basic density (BD), cellulose content (CC), hemicellulose content (HC), and lignin content (LC). Signals of strong differentially expressed genes (DEGs) and differentially spliced genes (DSGs) were detected in all four groups of wood properties, suggesting that gene transcription and selective splicing may have important regulatory roles in wood properties. We found that there was little overlap between DEGs and DSGs in groups of the same trait. Furthermore, the key DEGs and DSGs that were detected simultaneously in the four groups tended to be enriched in different Gene Ontology terms, Kyoto Encyclopedia of Genes and Genomes pathways, and transcription factors. These results implied that regulation of gene transcription and AS is controlled by independent regulatory systems in wood formation. Lastly, we detected transcript levels of known wood biosynthetic genes and found that 79 genes encoding mainly enzymes or proteins such as UGT, LAC, CAD, and CESA may be involved in the positive or negative regulation of wood properties. This study reveals potential molecular mechanisms that may regulate wood formation and will contribute to the genetic improvement of *Eucalyptus*.

**Keywords:** *Eucalyptus*; RNA splicing; transcription factors; wood formation; differentially expressed gene





## 1. Introduction

Widely planted in tropical and subtropical regions, *Eucalyptus* species are grown for timber, kraft pulp, and paper production due to their superior growth and adaptability to a wide range of sites [1]. Although there are over 700 species, just nine species are estimated to contribute over 90% of the planted area, either as pure species or as hybrid combinations. *Eucalyptus urophylla* (Timor white gum) and *E. tereticornis* (forest red gum) are both among the nine widely planted species, the latter being a close relative of *E. camaldulensis* (river red gum), and they are widely cultivated, both as pure species and as hybrid combinations, in several tropical areas of the world [2], providing strong, hard, and durable heartwood for construction and heavy engineering [3,4]. The *E. urophylla × tereticornis* hybrid has been widely cultivated in coastal areas of southern China due to its fast growth,

favorable wood properties, cold hardiness, and typhoon resistance [5–7]. Like other important industrial *Eucalyptus* species, *E. urophylla × tereticornis* is also targeted by genetic improvement programs focused on improving wood quality [8,9]. Significant genetic and genotype-by-environment variation in growth and wood properties have been detected in *E. urophylla × tereticornis* hybrids [6]. These effects generate substantial differentiation in the phenotypes of several wood properties. Wood physical properties including wood basic density (BD) and chemical properties such as cellulose content (CC), hemicellulose content (HC), and lignin content (LC) are fundamental traits for kraft pulp and paper production [10,11]. Wood properties in trees are shaped by a large number of genomic variants, many of which may regulate gene transcription or alternative splicing (AS) [12].

Wood formation including cell division and expansion, secondary wall formation, and programmed cell death are intricate developmental processes controlled by numerous gene families [13,14]. The regulation of wood properties by transcription of lignocellulose-related genes has been highlighted in *Eucalyptus*. For example, Shinya et al. [15] compared the transcript levels of genes involved in wood formation in two hybrid genotypes of *E. urophylla × grandis* with different lignin contents and found that a number of genes associated with monolignol biosynthesis may regulate *Eucalyptus* wood composition. Nakahama et al. [16] compared *E. urophylla × grandis* hybrids of different BD and found that most lignocellulosic biosynthesis-related genes showed a trend toward higher levels of transcription in high-BD trees. The genetic dissection of these important wood properties allows us to identify major genomic regions containing candidate genes associated with wood development.

AS produces multiple messenger RNA (mRNA) isoforms from the immature mRNA of a single gene via regulated usage of RNA splice sites by retaining or removing different exons and introns to increase the diversity of the transcriptome and proteome [17]. In plants such as *Arabidopsis thaliana* [18] and *Oryza sativa* [19], AS has been shown to have an influence on growth and development and environmental adaptation. Among forest trees, the few AS studies that have been carried out focused on *Populus*, e.g., *P. trichocarpa* [20] and *P. deltoides* [21]. Meanwhile, we still know very little about the AS variation pattern in *Eucalyptus*, particularly in relation to wood-forming processes. Recently, high-throughput RNA sequencing (RNA-seq) technologies have greatly enhanced our ability to obtain gene transcription profiles and identify AS events [17,22,23]. Xu et al. [12] used RNA-seq to compare the features and types of AS sites in *E. grandis* with those of *Populus*. Their study identified a large number of xylem genes exhibiting AS; these affected one-quarter of the highly expressed transcripts in these two species [12]. However, further research is required to explore the contribution and interaction of AS and gene transcription in wood formation.

In the present study, we performed RNA-seq on the developing xylem of *E. urophylla × tereticornis* hybrids with the aim of exploring (1) the features and profiles of gene transcription and AS in individuals with contrasting wood properties, (2) the differentially expressed genes (DEGs) and differentially spliced genes (DSGs) that play key roles in wood formation and their functions, and (3) the regulation by known wood biosynthetic pathway genes of wood properties in *Eucalyptus.* This study reveals the potential molecular mechanisms controlling the wood formation, which will contribute to the genetic improvement of *Eucalyptus*.

## 2. Results

### 2.1. Transcriptome Sequencing Quality

Seventeen libraries based on samples taken from individual *E. urophylla × tereticornis* hybrid trees with contrasting phenotypes were sequenced using RNA-seq. After processing the raw sequence data, between 19 and 47 million clean reads were obtained from each library (Table S1). The average sequence data above 7G was qualified, and the GC content was between 48% and 53%. Due to differences in sequencing quality, reads from individual libraries mapping to the *E. grandis* reference genome [24] ranged from 79.4% to 92.4%, with

an average of 85.5% (Figure 1e). A total of 490,520 transcripts (between 23,948 and 34,875 per library) were obtained by reference-guided transcriptome mapping.

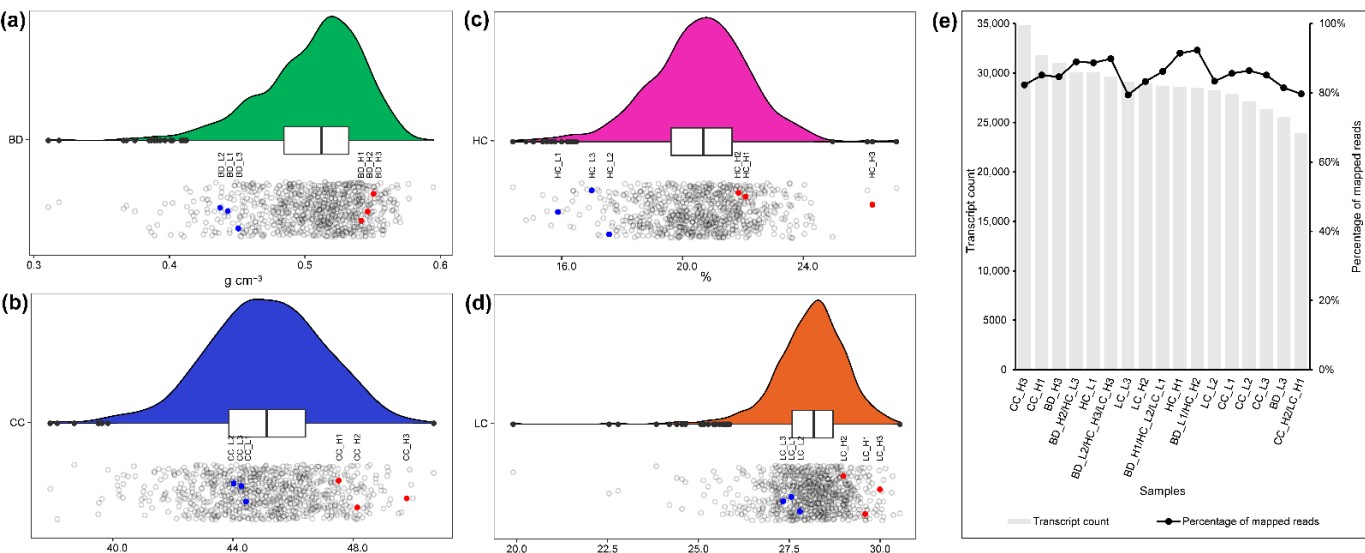

**Figure 1.** Variation of wood properties and selection of highly contrasting phenotypes of *E. urophylla × tereticornis* hybrids in an experimental population in southern China. Panels (**a**–**d**) show the distribution of phenotypic variation of BD (**a**), CC (**b**), HC (**c**), and LC (**d**) from 777 trees. The upper portion of panels a-d illustrate the approximately normal distribution of phenotypes in each trait (*x*-axis boxes indicate the interquartile range), while the lower proportion of panels are jittered scatterplots with the blue and red dots indicating three high (H) and three low (L) phenotypes selected for each trait. Panel (**e**) shows the sequence quality of RNA-seq performed on 17 samples, including transcript count and percentage of reads that mapped to the *E. grandis* reference genome. Note that the suffixes H and L indicate "high" and "low" phenotypes, with five trees representing two or three traits. For example, BD_L2/HC_H3/LC_H3 is a sampled tree that represents three traits (low BD, high HC, and high LC).

## 2.2. Analysis of Differentially Expressed Genes

Significant DEGs from four groups of contrasting wood properties were observed and visualized using volcano plots (Figure 2a). A total of 1258, 813, 4355, and 2521 genes were differentially expressed between the individuals with high and low phenotypes for BD, CC, HC, and LC, respectively, and the distribution of log$_2$ fold change (Log$_2$FC) ranged from −29.2 to 27.7 (Figure 2a and Table S2). The HC group showed a markedly stronger DEG signal than the BD, CC, and LC groups. In the BD group, the proportion of upregulated DEGs was higher than that of downregulated DEGs, while the converse was true for the CC, HC, and LC groups.

## 2.3. Alternative Splicing Identification and Differentially Spliced Gene Analysis

By analyzing the transcripts of individuals, 15,440, 10,853, 17,053, and 11,327 AS events in the BD, CC, HC, and LC groups were detected, respectively. After FDR correction, 771, 666, 1849, and 669 intron clusters were identified with significant differential splicing, of which the number of intron clusters with three introns was the largest in all groups (Figure 2b, Tables S2 and S3). An intron cluster was detected with 89 introns in the BD group, which was the splicing pattern involving the largest number of introns in all splicing events ("Df = 88, Cluster = chrChr02: clu_28141_NA"; Table S2). Furthermore, 649, 554, 1183, and 648 DSGs were found to be associated with the foregoing intron clusters in the BD, CC, HC, and LC groups, respectively, accounting for 1.8%, 1.5%, 3.3%, and 1.8% of *E. grandis* annotated genes (36,376). For each group, the number of genes that were both differentially expressed and differentially spliced ranged from 22 to 195 (Figure 3a and S1).

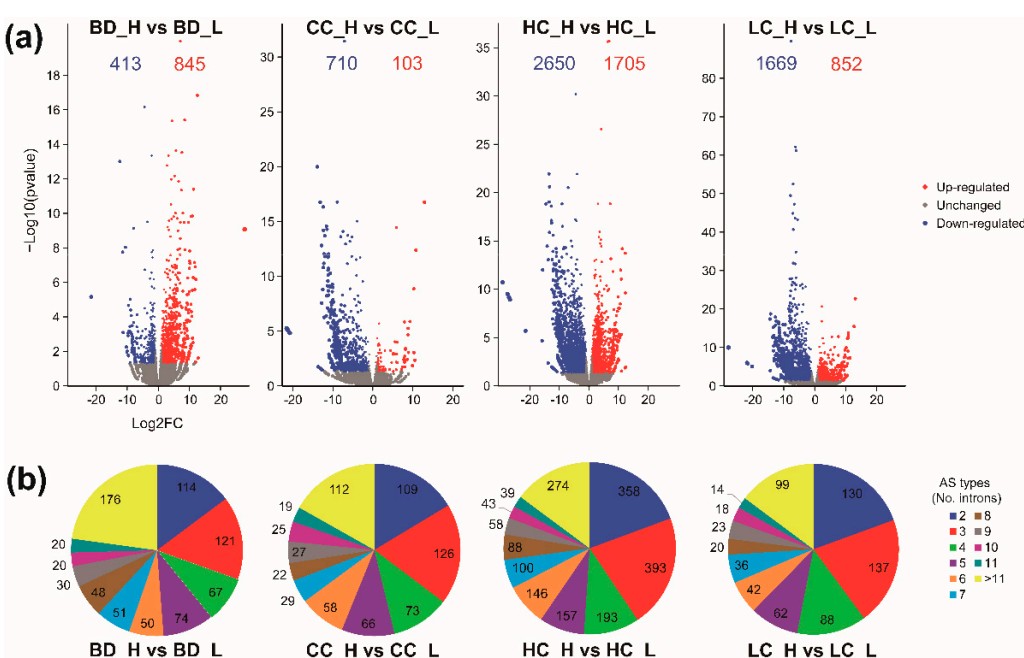

**Figure 2.** Identification of DEGs (**a**) and significant AS events (**b**) in groups of different traits. AS types represent the number of introns included in the AS events ranging from two to >11 introns, and the values in the circle represent the number of AS events.

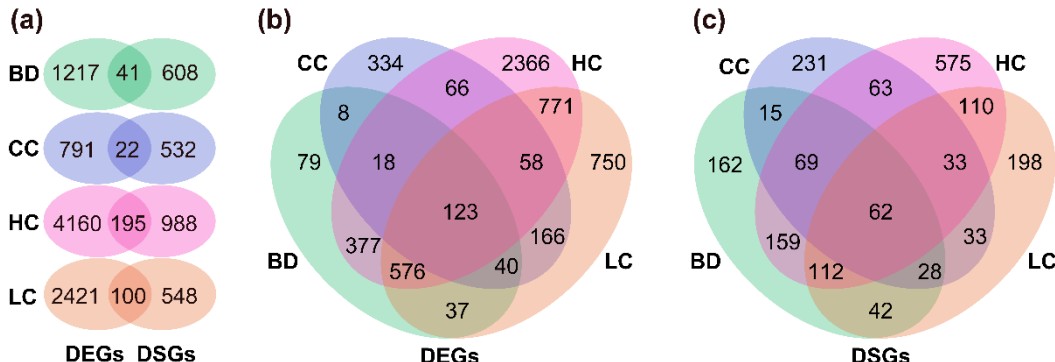

**Figure 3.** (**a**) Intersections between the DEGs and DSGs in groups of the same traits; intersections between (**b**) the DEGs and (**c**) DSGs in groups of different traits.

### 2.4. Functional Comparison of Key DEGs and DSGs

Intersections of the DEGs and DSGs among the four traits were summarized using Wayne diagrams (Figure 3), from which we identified 123 key DEGs and 62 key DSGs (Figure 3b,c and Table S4), i.e., they appeared to be acting on all four traits. We found that 123 key DEGs were significantly enriched in 62 Gene Ontology (GO) terms, including 46 biological processes, 13 molecular functions, and three cellular components (Figure 4a and Table S5). The functions of key DEGs were related to a number of GO terms, the most significant of which included basic cell functions, such as cell proliferation (GO:0008283), small molecule metabolic process (GO:0044281), and purine ribonucleotide biosynthetic process (GO:0009152), biological processes, including growth factor activity (GO:0008083), receptor binding (GO:0005102), phosphotransferase activity, and phosphate group as acceptor (GO:0016776), and molecular functions including integral component of membrane (GO:0016021), intrinsic component of membrane (GO:0031224), and membrane part (GO:0044425). Comparatively, we found that the 62 key DSGs were significantly enriched in a somewhat greater number (88) of GO terms (Figure 4b and Table S5). In the biological processes and molecular functions, the functions of key DSGs mainly related to Rab GDP-dissociation inhibitor activity (GO:0005093), ubiquitin protein ligase binding (GO:0031625),

ubiquitin-like protein ligase binding (GO:0044389), ubiquitin-dependent protein catabolic process (GO:0006511), modification-dependent protein catabolic process (GO:0019941), and modification-dependent macromolecule catabolic process (GO:0043632). In the cellular components, key DSGs were most significantly associated with cell (GO:0005623), intracellular (GO:0005622), and cell part (GO:0044464).

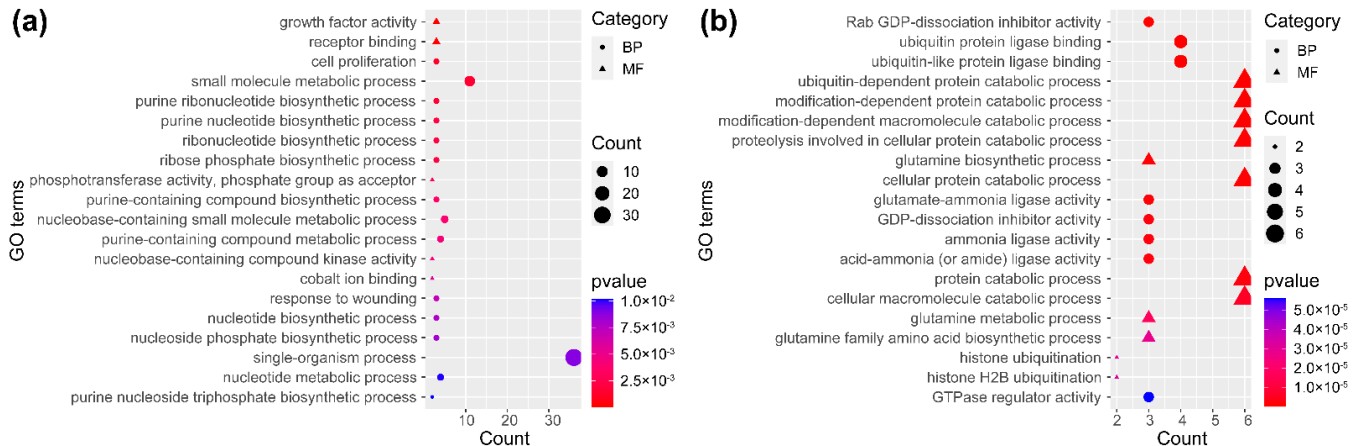

**Figure 4.** Top 20 terms of GO enrichment of key DEGs (**a**) and DSGs (**b**). BP and MF represent biological processes and molecular functions, respectively.

Kyoto Encyclopedia of Genes and Genomes (KEGG) pathway analysis was performed to explore the biosynthetic pathway in which the genes were involved. A total of 25 and 12 KEGG pathways were assigned from the key DEGs and DSGs, respectively, of which five and six pathways were significantly enriched (Table S6). The pathways significantly enriched by key DEGs were mainly related to metabolism, e.g., purine metabolism (ath00230), nitrogen metabolism (ath00910), and galactose metabolism (ath00052). The key DSGs, on the other hand, were mainly associated with ubiquitin-mediated proteolysis (ath04120), glyoxylate and dicarboxylate metabolism (ath00630), and protein processing in endoplasmic reticulum (ath04141). Additionally, other pathways not significantly enriched from key DSGs included phenylpropanoid biosynthesis (ath00940) and biosynthesis of secondary metabolites (ath01110), which may be related to various biological activities of wood cambium cells.

Furthermore, the key DEGs and DSGs were BLASTed against the transcription factors (TFs) of *E. grandis*. Among the key DEGs, a total of 21 TFs belonging to 14 families were identified, including HSF (three), MYB (two), NAC (two), and TCP (two) (Figure S2 and Table S7). Among the key DSGs, a total of 53 TFs were classified into 18 families that showed differentially spliced characteristics, of which the ERF family had the most abundant members (12), followed by bHLH (eight), HSF (four), Dof (four), etc. (Figure S2 and Table S7). The frequency of TFs identified in key DSGs was significantly higher than that of key DEGs. In several families, the number of TFs varied greatly between the key DEGs and DSGs, especially in ERF and bHLH. Both the key DEGs and the key DSGs contained some private TF families, e.g., DOG, BBR-BPC, and LBD; however, in several families they were consistent in number, e.g., MYB, NAC, and TCP. In general, the key DEGs and DSGs tended to perform different functions.

### 2.5. Wood Biosynthetic Pathway Genes

A total of 79 known wood biosynthetic pathway genes encoding 32 proteins or enzymes from 11 families were detected that may be involved in the regulation of contrasting phenotypes in *E. urophylla* × *tereticornis* hybrids (Figure 5 and Table S8). More than half (43) of the genes were associated with UDP-glucosyl transferase (UGT). Of these, *Eucgr.D02610* was upregulated in the BD group (Log$_2$FC = 7.9) and downregulated in the CC (−7.6), HC (−8.4), and LC (−8.3) groups; this gene may be involved in the regulation of all four

wood properties simultaneously. Differential expression of most genes was associated with HC, e.g., laccase (LAC) and cinnamyl-alcohol dehydrogenase (CAD). Increased CC may have been associated with upregulated expression of two cellulose synthase (CESA) genes. Similarly, downregulated expression of three and two CESA genes may have led to increases in HC and LC, respectively. Overall, these results suggest that the differential expression of these wood biosynthesis pathway-related genes may have led to corresponding phenotypic variation.

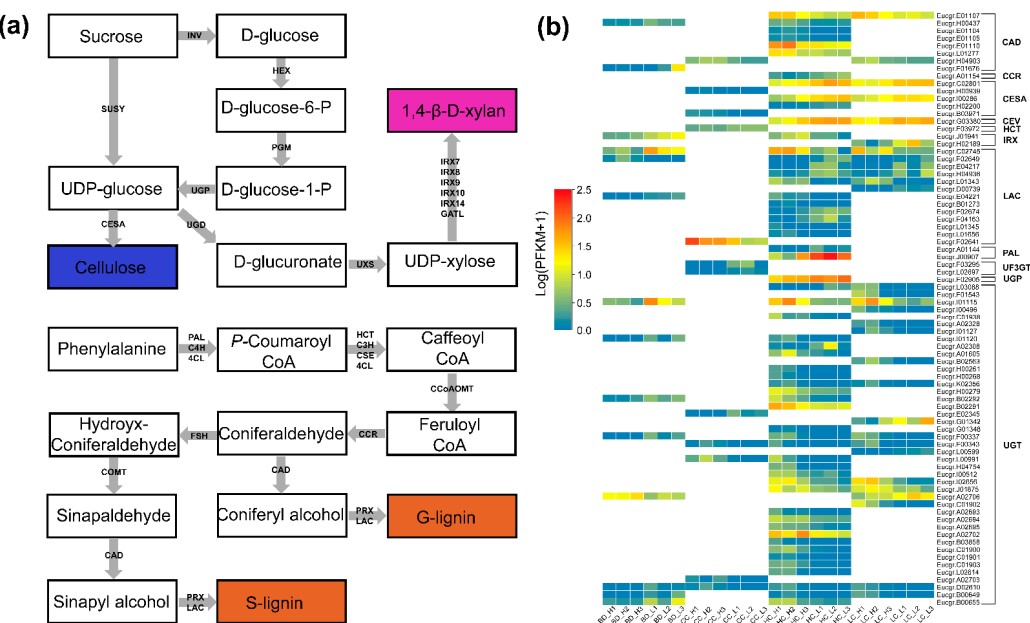

**Figure 5.** (**a**) Major biosynthesis pathway for cellulose, xylan, and lignin [15]. (**b**) Transcription of 79 wood biosynthetic pathway genes in groups of different traits. A redder color indicates higher gene transcription.

## 3. Discussion

### 3.1. DEGs and DSGs in Individuals with Contrasting Wood Properties

Signals of substantial gene transcriptional differences can often be detected in genotypes with highly divergent wood phenotypes, such as *E. urophylla* × *grandis* [15,16], *E. globulus* [25], *E. nitens* [26], and *Pinus radiata* [27]. These studies suggest that DEGs are important regulators of wood properties. Our study on *E. urophylla* × *tereticornis* wood traits contributes further evidence confirming the regulatory role of DEGs. The differences in gene transcription between high and low phenotypes were quantified using $\text{Log}_2\text{FC}$, and the results showed a range of $-21.6$–$27.7$ for the BD group samples, $-21.6$–$12.9$ for CC, $-29.2$–$12.5$ for HC, and $-27.9$–$13.1$ for LC (Figure 1a), indicating that considerable differences in gene transcription were detected in these groups. Compared to earlier studies in *Eucalyptus* [15,16,26,28], we found a greater number of xylem-related genes. This is probably due to the highly contrasting phenotypes we selected from within a larger breeding population with a large number of continuously variable phenotypes.

A better understanding of the AS patterns of genes associated with wood formation will provide a novel perspective on gene regulation of wood properties. Our results show that AS events in genes occur frequently in four groups of contrasting phenotypes of *E. urophylla* × *tereticornis* (Figure 1b). Splice junctions from two to 89 introns in our study and two to 61 exons in *P. deltoides* [21] indicate a remarkable diversity in AS types in forest tree species. Such abundance of AS sites and diversity of types suggest that AS is a universal regulation mechanism of wood formation. In principle, it is possible to summarize intron retention, skipped exons, 5′ and 3′ AS site usage, and additional complex events to identify AS events using Leafcutter [29]. The occurrence of AS detected in forest trees *P. trichocarpa* [20], *P. deltoides* [21], *P.* × *euramericana*, and *E. grandis* [12] were clearly

lower than those in the herbaceous plant *A. thaliana* [18] and in rice *O. sativa* [30]. The observed incidence of AS in genes in this *E. urophylla* × *tereticornis* study (666 to 1489 AS events; *Eucalyptus* genome size 653.98 Mb) was consistent with a study by Xu et al. [12] which considered *E. grandis* (2987 AS events). However, Xu et al. [12] also examined *P.* × *euramericana* (6031 AS events; *Populus* genome size 434.29 Mb) indicating that AS events appear to be more frequent in *Populus* than *Eucalyptus*, especially when taking into account the relatively small *Eucalyptus* genome.

Between 554 and 1293 DSGs in the four sets of phenotypes studied from significant AS events were identified, which only accounted for 1.5%–3.3% of the annotated genes in the *E. grandis* reference genome [24]. However, it is important to recognize that AS frequencies in plants are partially dependent on the annotation of genomes; as genome annotation improves, so too will the identification of AS sites. For instance, in *A. thaliana*, the observed AS frequency rose from 11.6% in a 2004 study [31] to 60% in 2012 as genome annotation improved [18]. With the ongoing development of next-generation sequencing, understanding the complexity and regulation of AS relating to wood formation will be made possible by employing larger and more comprehensively annotated samples for transcriptome sequencing. In the current study, the HC and CC groups exhibited the lowest (1.5%) and highest (3.3%) DSG frequencies, respectively, revealing differences in AS regulation of different wood property phenotypes in *Eucalyptus*.

### 3.2. The Relationship between DEGs and DSGs

In accordance with this study, little overlap between the DEGs and DSGs was found in previous studies of model and crop plants [19,32], suggesting that regulation of transcription and mRNA splicing is controlled by independent regulatory systems. This mechanism may be conserved in plants; for example, Xu et al. [12] found similar patterns in forest trees and observed even less overlap than model and crop plants [19,32]. This may be due to the influence of *cis* and *trans* regulation on the expression of specific genes. Evidence from a study of *P. deltoides* suggested that the expression of AS isomers was predominately *cis*-regulated, whereas splice junction usage was generally *trans*-regulated [21]. Likewise, we found few overlaps (22 to 195 genes) between the DEGs and DSGs in groups of the same trait, suggesting that independent regulatory mechanisms between transcription and mRNA splicing may be quite pronounced in forest trees. Moreover, functional analysis of the key DEGs and DSGs showed that their functions differed substantially. We found that the key DEGs mainly controlled the growth, metabolic, and biosynthesis processes, which are related to the metabolism and nutrient transport required for wood formation [33,34]. In contrast, the functions of the key DSGs were more usually associated with post-translational protein processing and modification modalities [35], such as ubiquitin protein ligase and proteolysis. The number of TFs identified within the ERF family was significantly higher than in other families, and these were mainly concentrated in the key DSGs. The ERF family reportedly regulates numerous biological and physiological plant processes, such as morphogenesis, physiological metabolism, and response mechanisms to various stresses [36–38]. Recently, Seyfferth et al. [39] identified 11 ERFs as putative regulatory hubs related to *Populus* wood formation, of which *ERF118* and *ERF119* were connected to xylem cell expansion and secondary cell wall formation. This suggests that key DSGs may be involved in the regulation of wood formation by encoding ERFs. In addition, even though the number of key DSGs was lower, they were enriched for GO terms, KEGG pathways, and TFs in higher numbers than key DEGs. This implies that DSGs may have additional regulatory functions. Indeed, a single gene can generate multiple mRNA variants through different AS patterns, which increases the diversity of genes or proteins [40]. We also found that DSGs may be more likely to be independent regulators of specific wood properties, as the number of private DSGs in each group of traits was higher than the other intersections (Figure 3c). Conversely, the intersection of the different groups observed in the Wayne diagram (Figure 3b) involved a higher number of DEGs, suggesting that the regulation of wood properties by DEGs exhibits pleiotropy. Collectively,

our results suggest that DEGs and DSGs tend to perform different regulatory functions during wood formation.

### 3.3. Transcriptional Levels of Wood Biosynthetic Pathway Genes

The important regulatory role of wood biosynthetic pathway genes on wood traits was previously demonstrated in forest tree species from the *Pinus* and *Eucalyptus* genera [16,27]. We, therefore, focused on the regulatory role of these genes for four important wood properties in *E. urophylla × tereticornis*. LAC is a class of glycoprotein polyphenol oxidases. LAC is involved in lignin biosynthesis by catalyzing the polymerization of two monolignols (sinapyl and coniferyl alcohol) [41]. In poplar, downregulated expression of two LAC encoding genes (*PtLAC2* and *PtLAC3*) was found to significantly affect xylem fiber cell morphology and lead to a reduction in LC [42]. Nakahama et al. [16] showed that one LAC gene was highly expressed, while two LAC genes were weakly expressed in genotypes of *E. urophylla × grandis* with high BD. Similarly, we found that, among the high BD phenotypes, one LAC gene was highly expressed, while the other LAC gene was weakly expressed. In addition, two LAC genes were highly expressed in the high LC group, and four were weakly or not expressed. This adds weight to the previous finding that the LAC gene has a significant effect on both the LC and the BD of eucalypts including *E. urophylla × tereticornis*. However, while Li et al. [27] found weak expression of the LAC gene but strong expression of the CESA gene in *P. radiata* with high BD, we found no difference in CESA gene expression between the high-BD and low-BD groups. This may be because BD is a complex physical trait integrating both chemical and structural determinants, which is, therefore, likely to involve numerous pathways of molecular regulation [43]. Shinya et al. [15] found that high LC may be associated with stronger expression of the CESA gene. The two CESA genes we found in the high CC phenotypes both appeared to be highly expressed, supporting the conclusions of Shinya et al. [15]. Interestingly, all three CESA genes found in the low HC phenotypes were highly expressed, implying that high expression of CESA may have contributed to the decrease in HC. Both cellulose and 1,4-β-D-xylan are in fact dependent on UDP-glucose for their synthesis, and high expression of CESA may promote greater conversion of UDP-glucose to cellulose, thereby reducing HC. Interestingly, UGT is a key class of enzymes for the synthesis of 1,4-β-D-xylan, whose glycosyl donor is UDP-glucose [44]. The variation between transcript levels of xylan biosynthetic pathway genes and HC content has not been documented in two previous studies of *Eucalyptus* [15,16]. However, we found that 32 of the 43 UGT genes were differentially expressed in the HC group and, therefore, may have contributed to the observed variation in HC.

### 3.4. Further Research Arising from This Study

Although we identified several DEGs and DSGs in xylem samples taken from phenotypes contrasting in key wood property traits, we acknowledge some difficulties in interpreting the results from this and other studies using similar methods (e.g., [12,15,16,20,21]). Forest trees are long-lived organisms, and wood-forming processes are known to vary seasonally and throughout the life of the tree [45]; a small proportion of xylem genes may even exhibit circadian variation [46]. Sampling tissue and assaying the transcriptome at a single timepoint is, therefore, effectively only giving a "snapshot" that is less likely to account for trait-forming processes than would a single sample in a shorter-lived species such as an annual crop. Wood traits, on the other hand, are typically sampled from one or multiple tree rings, effectively integrating the temporal variance in the process of wood formation. Associating a point sample representing transcriptome activity with a wood chemical sample that has formed over a much longer time period may, therefore, be problematic. To address this issue, it may be necessary to carry out a sequence of samples at different timepoints, both within years and throughout the life, or in plantation species, the crop rotation, of the trees. While this would be expensive and time-consuming to carry out in a single experiment, it may prove to be possible to build up a picture of whether some

key genes are actively associated with wood trait phenotypic status at different timepoints by combining data from different studies. Studies that have attempted to include temporal variance include Zhang et al. [33], who compared the transcriptomes of *Cunninghamia lanceolata* xylem sampled at three cultivation ages (7, 15, and 21 years), and Chao et al. [47], who compared the transcriptomes of *Populus* stemwood sampled from six developmental zones. Significantly, they both found that some DEGs were involved in the regulation of wood formation or secondary growth at multiple timepoints. It is also promising that certain DEGs and DSGs identified in this study appear to have also been identified in other studies of eucalypts and other species. This is despite the fact that the RNA involved in these separate studies was sampled at single timepoints, most usually in the season of active tree growth, but at different ages and probably different developmental stages in the plantation life cycles. Identification of point-sampled transcriptomic processes that can reliably be associated with wood property traits will be an important practical research application in the future.

## 4. Materials and Methods

### 4.1. Plant Materials

The experimental plantation located in Gonghe town, Heshan city, Guangdong province, China (22°32′ N, 113°02′ E) comprised 320 clonal genotypes of *E. urophylla × tereticornis* hybrid established as rooted cuttings. Each clone comprised four ramets or fewer, with a total of 777 trees (Figure 1a–d). Wood properties of each clonal ramet were assessed at age 8 years. Traits measured included BD (g·cm$^{-3}$), CC (%), HC (%), and LC (%) using near-infrared (NIR) analysis (Figure 1a–d) using the method as detailed by Yang et al. [6]. Three individual tree replicates were selected to represent the two contrasting high and low phenotype groups for each of the four traits (BD, CC, HC, and LC) implying a total of 24 samples. In fact, five selected individuals were representative of more than one trait (Figure 1a–e), resulting in samples being taken from 17 trees which covered all 24 combinations (three high and three low phenotypes each, from each of four traits). The xylem/cambium tissues of the 17 individuals were collected in mid-July, which is in the active period for tree growth in Guangdong, China. Tissue samples were harvested at breast height (approximately 1.30 m) from the stem's south-facing aspect. Samples were harvested by removal of a bark window (a square of about 10 cm × 10 cm), scraping developing xylem tissues, and removing the phloem. All samples were collected between 9:00 and 11:00 a.m. on the same day and immediately frozen in dry ice. The samples were then transferred to a −80 °C refrigerator for storage.

### 4.2. RNA Extraction, Library Construction and RNA Sequencing

Total RNA was extracted using the EASYspin Plus Plant RNA Kit (Aidlab, Beijing, China). RNA quality score (RQS) was evaluated using Labchip GXII Touch (Perkin Elmer, Waltham, MA, USA). By using VAHTS$^{TM}$ Stranded mRNA-seq Library Prep Kit (Vazyme Biotech, Nanjing, China), the extracted RNA was prepared for the mRNA-seq library. The mRNA-seq library size and concentration were assessed using the Labchip GXII Touch and the QUBIT fluorometer 3.0 (ThermoFisher, Waltham, MA, USA). The constructed mRNA-seq library was sequenced using the Nextseq-500 platform (Illumina, San Diego, CA, USA).

### 4.3. Quality Control of Sequences and Reference-Guided Transcriptome Mapping

Raw data were converted to FASTQ format using the bcl2fastq2 tool (https://support.illumina.com/sequencing/sequencing_software/bcl2fastq-conversion-software.html, accessed on 1 March 2018). The raw data were then filtered by removing adaptors using Trimmomatic-0.36 [48]. To obtain high-quality, clean reads for mapping, raw sequencing reads were filtered by FastQC, including removing reads with homopolymer counts >20% and Poly A counts >20%. In addition, reads with sequence quality (>20) and <5 missing values were retained. Following this quality control step, the clean reads from each

sample were mapped to the *E. grandis* reference genome (https://phytozome.jgi.doe.gov/pz/portal.html, accessed on 1 June 2022; *Eucalyptus grandis* v2.0) with HISAT2 [49]. The resulting alignments were processed using StringTie [50] for transcript assembly, estimation of the expression levels of each gene, and obtaining the gene transcription matrices.

*4.4. Differentially Expressed Gene and Differentially Spliced Gene Detection*

Differential gene expression analysis was carried out using the R package DESeq2 [51]. Differentially expressed genes (DEGs) were defined as those with $Log_2FC \geq 1$ and a false discovery rate (FDR)-corrected $p \leq 0.05$. Volcano, Venn, and heat maps were drawn using TBtools v1.0683 [52]. AS events were quantified in both BD, CC, HC, and LC samples using Leafcutter [29], which is an annotation-free method based on exon–exon junction reads to identify the excised introns. Leafcutter defines introns that overlap and share the acceptor or donor splice site as intron clusters, and it summarizes AS events on the basis of intron excision differences. We identified differentially spliced genes (DSGs) from significant alternatively spliced intron clusters with a threshold of FDR-corrected $p \leq 0.05$.

*4.5. Analysis of Gene Ontology and Biological Pathways and Transcription Factor Prediction*

We focused on key DEGs and DSGs that were identified simultaneously in four trait groups and analyzed their function. Firstly, DEGs and DSGs were aligned to *A. thaliana* with an *E*-value $\leq 10^{-5}$. Orthologous genes from *A. thaliana* identified in *E. grandis* were then used to perform the downstream analysis [53]. Then, GO enrichment analysis classified the DEGs and DSGs of each group according to cellular component, molecular function, and biological process using the PlantRegMap [54] with a threshold of FDR-corrected $p \leq 0.05$, and KEGG pathway analysis using Kobas v3.0 with the same $p$-value [55]. In addition, the 2163 TFs from *E. grandis* collected in the Plant TF Database (PlantTFDB) were used as references; then, genes as targets of TFs were predicted using PlantTFDB v4.0 [56] from DEGs and DSGs of each trait, respectively. In addition, we investigated the transcript levels of known cellulose, hemicellulose, and lignin biosynthesis pathway-related genes in all DEGs [15].

**5. Conclusions**

We investigated DEGs and DSGs in *E. urophylla* × *tereticornis* hybrid phenotypes with highly contrasting BD, CC, HC, and LC. The key DEGs and DSGs detected simultaneously in the four groups of wood properties tend to play different regulatory functions during wood formation. Seventy-nine wood biosynthetic genes may be involved in the up- or downregulation of wood properties in *E. urophylla* × *tereticornis* hybrids. Our findings generally concur with several findings from other studies in model plants, agricultural crops, and forest tree species and contribute to the growing understanding of the role of specific genes, gene families, and their regulation in the development of wood traits of economic importance.

**Supplementary Materials:** Supplementary materials for this study can be found at https://www.mdpi.com/article/10.3390/f13071102/s1. Raw RNA-seq data are available at NCBI, accession number: PRJNA705012. Figure S1. The genes that were both differentially expressed and differentially spliced in groups of the same traits. Figure S2. Transcription factor prediction of key DEGs and DSGs. Table S1. Summary of transcriptome sequencing with highly contrasting phenotypic individuals. Table S2. List of DEGs and DSGs. Table S3. Information of significant intron splicing. Table S4. The genes that were both differentially expressed and differentially spliced in groups of the same traits. Table S5. List of GO terms of key DEGs and DSGs. Table S6. List of KEGG pathways of key DEGs and DSGs. Table S7. List of TFs of key DEGs and DSGs. Table S8. Information of 79 wood biosynthesis pathway genes.

**Author Contributions:** Conceptualized and designed the experiments, and finalized the paper, F.L.; performed the experiments and analyzed the data, X.Z., J.H. and C.Z.; participated in the measurement of wood chemical properties and collected samples, Q.W. and S.C.; wrote the manuscript with

assistance from all other authors, X.Z., J.H., D.B. and F.L. All authors read and agreed to the published version of the manuscript.

**Funding:** This work was financially supported by the Fundamental Research Funds of Chinese Academy of Forestry (CAFYBB2021ZA001), Guangdong Natural Science Foundation (2020A1515010974), and Ministry of Science and Technology of China (2016YFD0600101).

**Institutional Review Board Statement:** Not applicable.

**Informed Consent Statement:** Not applicable.

**Data Availability Statement:** Not applicable.

**Acknowledgments:** We thank Siming Gan and his colleagues for their coordination of the research project, and for the management and cultivation of clonal materials. We also thank Yaqin Wang and Li Wang for kind assistance in RNA extraction and RNA-seq library construction.

**Conflicts of Interest:** The authors declare no conflict of interest.

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
