# Peer review of "Xylem Transcriptome Analysis in Contrasting Wood Phenotypes of Eucalyptus urophylla × tereticornis Hybrids"

_forests, doi:10.3390/f13071102_

Round 1
Reviewer 1 Report
Dear Authors,
Thank you very much for providing a very detailed point to point response to my comments.
I think that you convincingly answered to many of the questions that I raised in my previous round of review. Text has been improved.
Unfortunately, I still think that the approach used is not adequate to disentangle the biological processes that are the declared as aim of the study.
Furthermore, some indispensable phenological/anatomical observations are still lacking; such data can not be provided now without performing a new experiment.
Regardless of the tangible improvements, I am sorry to conclude that the paper in my opinion can not be accepted for publication.
Best wishes
Author Response
We thank this reviewer for their comments and suggestions. We regret that we could not add experiments related to phenological/anatomical observations to this paper. It will be considered in our subsequent studies. Thanks again for the review of our manuscript.

Reviewer 2 Report
I found this manuscript to be very interesting. The design has some weaknesses due to potential differences in developmental stages etc but the authors acknowledge this and have left additional experiments for the future. At the same time, I suggest some RT-PCR with some of the differentially expressed genes so they can confirm the results in a greater number of trees. It might be the fastest and cheapest way to see if slight differences in developmental stage have an effect too.
Here are a few little things I noticed that could be improved:
In the abstract (line 22), I feel that gene transcription might be an incomplete term since they are measuring levels of RNA and that is a combination of transcription and stability. Maybe the sentence could be changed to ... transcript abundance and... or "gene transcription and stability... or something along those lines.
Figure 1 (line 109) Pane e should be panel e
Line 116 - 1,258, 813, 4,355 and 2,521 genes... In scientific writing, commas are often omitted in four-digit numbers and that would make the line more readable. 1258, 813, 4355 and 2521 genes.
Sometime an additional word would make the paper a little better. For example, line 119 - HC group showed a ... Personally, I think "The HC group..." would be better. Same later in the sentence where I would prefer "the BD, CC, and LC groups". There are other places like that. The editorial office may fix minor things like that.
Author Response
Point 1: Comments and Suggestions for Authors
I found this manuscript to be very interesting. The design has some weaknesses due to potential differences in developmental stages etc but the authors acknowledge this and have left additional experiments for the future. At the same time, I suggest some RT-PCR with some of the differentially expressed genes so they can confirm the results in a greater number of trees. It might be the fastest and cheapest way to see if slight differences in developmental stage have an effect too.
Response1: We focused on some specific functional genes that may affect wood phenotypes. We agree that performing a validation would be desirable. However, since our experiments were completed a long time ago, the previously preserved RNA would now be degraded. Performing RT-PCR with new experimental material will take a significant period of time. It would be necessary to re-investigate wood properties, extract RNA and design primers, etc. This is a quite long cycle, so, regretfully we cannot provide the results of RT-PCR. We also note that other studies have not included RT-PCR results, but do provide some meaningful results [1–3].
Point 2: Here are a few little things I noticed that could be improved:
In the abstract (line 22), I feel that gene transcription might be an incomplete term since they are measuring levels of RNA and that is a combination of transcription and stability. Maybe the sentence could be changed to ... transcript abundance and... or "gene transcription and stability... or something along those lines.
Response2: We measured the transcript level of the RNA, but we compared it to the reference genome of the Eucalyptus grandis. Therefore, we were concerned with the transcript levels of particular genes, such as Eucgr.XXXXXX (the gene number of mapped E. grandis gene). It is also referred to as gene transcription in previous studies [4,5] and, in addition, as gene expression [6,7], the latter being the term we used previously.
Point 3: Figure 1 (line 109) Pane e should be panel e
Response3: We have made corrections.
Point 4: Line 116 - 1,258, 813, 4,355 and 2,521 genes... In scientific writing, commas are often omitted in four-digit numbers and that would make the line more readable. 1258, 813, 4355 and 2521 genes.
Response4: We have revised the numbers of all four-digit numbers in the manuscript.
Point 5: Sometime an additional word would make the paper a little better. For example, line 119 - HC group showed a ... Personally, I think "The HC group..." would be better. Same later in the sentence where I would prefer "the BD, CC, and LC groups". There are other places like that. The editorial office may fix minor things like that.
Response5: We have revised the above mentioned places mentioned by the reviewers as well as other places in the text.
References
- Gil-Muñoz, F.; Delhomme, N.; Quiñones, A.; Naval, M. del M.; Badenes, M.L.; García-Gil, M.R. Transcriptomic Analysis Reveals Salt Tolerance Mechanisms Present in Date-Plum Persimmon Rootstock (Diospyros Lotus L.). Agronomy 2020, 10, 1703, doi:10.3390/agronomy10111703.
- Shen, H.; He, H.; Lu, C.; Liang, Y.; Wu, H.; Zheng, L.; Wang, X.; Liang, G. Comparative Transcriptome Analysis of Two Populations of Dastarcus Helophoroides (Fairmaire) under High Temperature Stress. Forests 2021, 13, 13, doi:10.3390/f13010013.
- Swanepoel, S.; Oates, C.N.; Shuey, L.S.; Pegg, G.S.; Naidoo, S. Transcriptome Analysis of Eucalyptus Grandis Implicates Brassinosteroid Signaling in Defense Against Myrtle Rust (Austropuccinia Psidii). Front. For. Glob. Change 2021, 4, 778611, doi:10.3389/ffgc.2021.778611.
- Xiang, R.; Hayes, B.J.; Vander Jagt, C.J.; MacLeod, I.M.; Khansefid, M.; Bowman, P.J.; Yuan, Z.; Prowse-Wilkins, C.P.; Reich, C.M.; Mason, B.A.; et al. Genome Variants Associated with RNA Splicing Variations in Bovine Are Extensively Shared between Tissues. BMC Genomics 2018, 19, 521, doi:10.1186/s12864-018-4902-8.
- Zhao, X.; Yu, L.; Liu, Z.; Liu, J.; Ji, X.; Zhang, X.; Liu, M.; Mei, Y.; Zeng, F.; Zhan, Y. Transcriptome Analysis for Fraxinus Mandshurica Rupr. Seedlings from Different Carbon Sequestration Provenances in Response to Nitrogen Deficiency. Forests 2021, 12, 257, doi:10.3390/f12020257.
- Yan, S.; Zhang, D.; Chen, S.; Chen, S. Transcriptome Sequencing Analysis of Birch (Betula Platyphylla Sukaczev) under Low-Temperature Stress. Forests 2020, 11, 970, doi:10.3390/f11090970.
- Zhang, Y.; Li, J.; Li, C.; Chen, S.; Tang, Q.; Xiao, Y.; Zhong, L.; Chen, Y.; Chen, B. Gene Expression Programs during Callus Development in Tissue Culture of Two Eucalyptus Species. BMC Plant Biol 2022, 22, 1, doi:10.1186/s12870-021-03391-x.

This manuscript is a resubmission of an earlier submission. The following is a list of the peer review reports and author responses from that submission.
Round 1
Reviewer 1 Report
Authors used transcriptome analysis and tried to clarify the relationship among alternative splicing, gene expression, and phenotypic variation in wood properties of Eucalyptus urophylla ´ E. tereticornis hybrids. Seventeen libraries were sequenced and differentially expressed genes were analyzed. The information provided in this manuscript is not sufficient to meet the goal of this study.
For plant materials, individuals grouped into wood basic density (BD), cellulose content (CC), hemicellulose content (HC), and lignin content (LC) should be indicated in Table S1.
To broaden the knowledge of the wood formation and contribute to the further screening of key genes in the process of wood biosynthesis, the list of overlapping genes between the four wood properties is much more important than the abundance of DEGs and DSGs in Figure 3~5.
Expression profiles of genes involved in cellulose, hemicellulose, and lignin biosynthesis should be indicated to ensure the relationship between gene expression variation and wood formation and to screen candidate key genes for wood formation.
Reviewer 2 Report
Overall, I found this to be a very interesting topic. The results were not as interesting as I anticipated based on the abstract but only because it is written as a bioinformatics type paper giving lots of statistics and I'm more interested in the molecular biology of what is going on. I could have used a little more help understanding some of the figures. It is probably very clear to people working in the field but to somebody very interested but not in the field, a simple explanation would have helped me. I'm not saying it needs to be added, just that it would have made me enjoy the paper more. I'm not at all surprised by the number of differentially expressed genes but I do find it very interesting how many differentially spliced genes were found. I'd be interested in more information on how they changed the proteins. Maybe in the next manuscript? As a molecular biologist with interests in wood development, I was particularly interested in the part on transcription factors and hope this work will be expanded.
The manuscript was very well written. I have read a lot of manuscripts for Forests and this was one of the best. I did notice a few tiny grammar or spelling errors and am listing them by line number so the authors or editors can fix them. I'm sure I missed some.
72 - these affected around quarter (should be one quarter or a quarter)
204 - the growth and developmental process (should be processes)
278 - There is important (It is important)
320 - by using E. grandis reference genome (the E. grandis reference)
345 - ndividual (individual)
Reviewer 3 Report
The paper from Zhu et al. presents a study of differential mRNA accumulation and differential post-transcriptional regulation in E. urophylla × E. tereticornis individuals showing contrasting wood traits.
The paper addresses an interesting topic, with a strong relevance both from a theoretical and applied research point of view.
The molecular analyses were conducted with appropriate methodologies and data analyses tools.
Nevertheless, I have serious concerns about the theoretical framework and experimental approach used and (see below).
Another very crucial point is that I think that the paper needs a wide text revision as there are throughout the manuscript many terminology inaccuracies, that may be interpreted in some cases as conceptual errors.
Please provide a point-to-point response to the comments.
Main comment:
The paper uses a set of contrasting phenotype individuals from a population of E. urophylla × E. tereticornis clones that were characterized for some wood traits (using NIR) by Yang et al., (2018).
Wood formation is a complex process, divided in different phenophases spanning from onset of cambial activity (spring) to latewood formation (lignification, cell death) spanning the whole growing season, even after the end of cambial cell division (so even in autumn). Indeed, it is a process driven by endogenous as well as external environmental factors regulating cambial phenology. Wood traits are measured on fully formed tree rings that are the result of multi-year biological processes; I think that using just one time point analysis of the very transient mRNA populations is not really the best approach to build association with wood traits under normal growing conditions (so in absence of specific manipulations affecting wood formation).
In my opinion, what it is studied in the submitted paper is a snapshot of differential mRNA accumulation or AS in (unspecified) developing tissues of a set of different genotypes, ad even if this is interesting and represents a basis for future research, any association with wood traits phenotypes is at this stage purely speculative.
It would have represented a more significative contribution to validate differential mRNA accumulation or AS of some genes from the RNAseq DEGs in controlled experiments and associate them with measurements of phenological, anatomical and chemical properties during the wood formation process.
Furthermore, a main issue that should be properly addressed is the lack of phenological characterization of the wood forming process; the Authors only declare (lines 348-350) that sampling occurred in “mid- July which is in the active period for tree growth in Guangdong, China”. In which phase of the wood formation process were the plant (e.g., by means of microscopy)? Probably even more importantly, were all the genotypes in the same phase? These are basic data to build associations, as well as biochemical data on the same samples used for RNAseq. The tissue/s used for analysis must be more precisely described (xylem only, xylem-cambial region-phloem, etc.).
Specific comments:
Title: In my opinion the title shows a conceptual error that is repeated in the text: it is not possible to study gene expression or AS (even accepting this misleading term, often used) on wood properties because “wood properties” is not a biological process. See also main comment above.
Please use italics for species scientific names (check all manuscript).
Abstract sentence at lines 19-22. The paper does not study gene expression, but rather mRNA accumulation (the two terms are NOT synonyms!). The sentence is quite hard to read, please rephrase it for clarity.
Abstract sentence at lines 22-24. The meaning of “a total of 813 to 4,355 differentially expressed….” Is not very clear and probably is not very meaningful to report the number of DEGs and DSGs in the abstract. Please rephrase for clarity.
Abstract sentence at lines 28-30. It is not a conclusion of this manuscript that differential mRNA accumulation and post-transcriptional regulation act to determine contrasting phenotypes (including wood traits) but is rather a widely accepted concept. Please delete the sentence.
Lines 55-56. Transcriptions factors and protein kinase do not belong to “biological pathways”, please rephrase for clarity.
Line 65: ….and some angiosperms. Arabidopsis, Oryza, etc. are also angiosperm, please correct.
Lines 80-82. It is important to give a quick explanation of this parameters, especially if their variation is proportional, considering that some of the genotypes used are representative of more than 1 trait.
Line 83: ….by using RNA-Seq. This might be moved upper in the sentence for more clarity.
Lines 83-84. "However, improving wood quality remains a major challenge in Eucalyptus”. This sentence seems disconnected from the context, please check.
Line 96. Please provide an explanation of the different mapping results to E. grandis genome.
Lines 102-103. Please be consistent in using full variable names or abbreviations.
Lines 110-111. The sentence “DEGs of four wood properties” is meaningless, please correct.
Lines 115-117: This is a normal result (see also comments for lines 247-248), as differences (if any) in the physiological/phenological, etc. state of the experimental plants are NOT presented in this work, so no strong differential mRNA accumulation is expected.
Lines 118-121. See comment for lines 237-241.
Line 204. TFs are involved in ALL processes and ALL organisms, not only growth and development in woody plants. Please delete the sentence.
Lines 205-206. I think that the concept is widely known and should be eliminated.
Lines 206-207. as above.
Lines 113-114. See comment for lines 241-245.
Line 137. “transcripts of wood properties” is meaningless, please rephrase.
Lines 152-153. There is no demonstration that DEGs and DSGs in this work are actually related to wood properties. Any association is purely speculative. Please clarify and rephrase.
Lines 231-232. Paragraph title and following text: differentially expressed gene do not specifically regulate the wood formation process, but rather the differential expression of some (or many) genes regulates any process leading to all different physiological (and then phenotypic) response. Furthermore, the concept is obvious. I suggest changing the title and delete the first part of the sentence or rephrase it.
Lines 235-237. Please clarify what is intended for “clear DEGs gap”. Which is the biological meaning of different number of up or down regulated mRNA accumulation?
Lines 237. Again: “wood property gene expression” has no meaning, please correct.
Lines 237-241. I do not understand which is the meaning of comparing logFC with other studies, as these values are related to experimental specific conditions and are NOT comparable. Please clarify.
Lines 241-245. Which is the biological meaning of logFC change? Please provide an hypothesis otherwise is just a description of sequencing technical results.
Lines 247-248. It is normal that only a small proportion of genes shows differential mRNA accumulation, especially because the authors are studying genotypes variation and not plants showing clear and documented physiological differences at the sampling date.
Lines 255-256. Again, this is speculative, not related to wood properties, but related to wood formation in different genotypes (which phase of wood formation is not known).
Lines 314-315. As above, TFs regulate all processes, not just wood formation please change paragraph title and rephrase the following sentence.
Lines 344-345. Please check the sentence.
Lines 358-359. the two sentences seem in the wrong order, please check.
Conclusions: again, DEGs and DSGs identified are not related to wood properties, please rephrase